# Genetic Complexity in Spondyloarthritis: Contributions of HLA-B Alleles Beyond HLA-B*27 in Romanian Patients

**DOI:** 10.3390/ijms26157617

**Published:** 2025-08-06

**Authors:** Ruxandra-Elena Nagit, Mariana Pavel-Tanasa, Corina Cianga, Elena Rezus, Petru Cianga

**Affiliations:** 1Immunology Department, “Grigore T. Popa” University of Medicine and Pharmacy, 700115 Iași, Romaniamariana.pavel-tanasa@umfiasi.ro (M.P.-T.); corina.cianga@umfiasi.ro (C.C.); 2Immunology Laboratory, “St. Spiridon” Clinical Hospital, 700106 Iași, Romania; 3Rheumatology Department, “Grigore T. Popa” University of Medicine and Pharmacy, 700115 Iași, Romania; elena.rezus@umfiasi.ro; 4Rheumatology Clinic, Clinical Rehabilitation Hospital, 14 Pantelimon Halipa Street, 700661 Iasi, Romania

**Keywords:** spondyloarthritis, HLA-B*27-negative, immunogenetics, Romanian cohort, cross-reactivity

## Abstract

This study looked at genetic factors in a group of inflammatory joint diseases called spondyloarthritis, focusing on patients from Northeastern Romania who do not carry a common genetic marker known as HLA-B*27. These patients are often underrepresented in research since little is known about the genetic risk factors and disease mechanisms affecting HLA-B*27-negative patients, especially in Eastern European groups. We analyzed 263 individuals and compared their genetic profiles to those of healthy people. Certain gene variants, such as HLA-B*47 and HLA-B*54, were more common in patients, while others were less frequent. Some variants were linked to specific symptoms like skin disease or inflammation. A group of genes, including HLA-B*08, B*18, and B*35, showed similar peptide-binding repertoires and may work together to increase disease risk. These findings suggest that spondyloarthritis in people without HLA-B27 may develop through different genetic pathways, and understanding these differences can improve diagnosis and support the development of more personalized treatments. This study also emphasizes the need to include underrepresented populations in genetic research to better reflect global diversity in disease.

## 1. Introduction

Spondyloarthritis (SpA) encompasses a group of chronic immune-mediated disorders characterized by inflammation of the axial skeleton, peripheral joints, and entheses, often accompanied by extra-articular manifestations such as anterior uveitis, psoriasis, or inflammatory bowel disease. Based on clinical presentation and imaging findings, SpA is classified into axial and peripheral forms (axSpA and pSpA, respectively) [1]. AxSpA includes both non-radiographic disease (nr-axSpA) and radiographic axial SpA—also known as ankylosing spondylitis (AS)—in which sacroiliitis is visible on conventional radiographs [2]. Peripheral forms include psoriatic arthritis (PsA), reactive arthritis (ReA), enteropathic arthritis, and undifferentiated SpA (USpA) [3].

These clinical subtypes share not only overlapping manifestations but also a common immunogenetic background. The strongest genetic association is with the HLA-B*27 allele, found in up to 95% of patients with AS. However, its contribution accounts for only approximately 20% of the estimated heritability [4,5]. Yet, these findings are largely derived from Western European, North American, and East Asian cohorts, with Eastern European populations remaining significantly underrepresented. Given the population-specific variation in HLA allele frequencies, this underrepresentation may obscure regionally relevant associations and limit translational applicability.

A distinct line of inquiry emerged from technical challenges observed during flow cytometric detection of HLA-B27, where unexpected cross-reactivities suggested structural homology with other HLA-B molecules, like HLA-B12, B37, B40, B41 [6], HLA-B07, HLA-B18, B35, B37, B38, and B39 [7]. These findings laid the conceptual groundwork for exploring whether cross-reactive HLA-B antigens might also contribute to SpA susceptibility, particularly in HLA-B27-negative individuals.

## 2. Results

### 2.1. Demographic and Clinical Characteristics

A total of 263 HLA-B*27-negative patients meeting the ASAS classification criteria for axial or peripheral SpA were analyzed. The cohort exhibited a female predominance (153 females vs. 110 males). The mean age at diagnosis was 39.3 years, with women diagnosed slightly later on average (40.35 years) compared to men (37.42 years). The average diagnostic delay was 3.6 years, with no substantial difference between sexes (3.61 years in women and 3.66 years in men). Patients with inflammatory bowel disease experienced the longest diagnostic delay (4.3 years), while those with uveitis were diagnosed earlier (1.5 years). Notably, 88 individuals (33.5%) had previously received a diagnosis of nonspecific low back pain, which contributed to diagnostic delay.

Morning stiffness had a mean duration of 43 min across participants. Lumbar mobility was reduced in the majority of patients, with a mean Schober test value of 2.95 cm (normative threshold: ≥5 cm), observed in 208 individuals. Inflammatory low back pain was reported by 184 patients, and peripheral involvement was documented in 185 cases. Lower limb symptoms were more common (157 cases) than upper limb symptoms (118 cases).

Specific peripheral manifestations included peripheral arthritis in 107 patients, enthesitis in 41, dactylitis in 15, and heel pain in 47. Extra-articular features included psoriasis (*n* = 33), inflammatory bowel disease (*n* = 7), and uveitis (*n* = 25), the majority being anterior (*n* = 15), followed by intermediate (*n* = 8) and posterior (*n* = 3) subtypes. Elevated inflammatory markers (CRP and/or ESR) were present in 78 individuals.

Radiological evidence of sacroiliitis (grade ≥ 2) was identified in 90 patients, while 108 exhibited grade 1 sacroiliitis or MRI-detectable structural/active lesions. Bilateral radiographic involvement was observed in 118 cases.

A summary of clinical, laboratory, and imaging data is provided in Table 1.

### 2.2. HLA-B Allele Distribution in HLA-B*27-Negative SpA Patients

In the analyzed cohort, six alleles accounted for the majority of HLA-B variation: HLA-B*35 (35%), HLA-B*18 (28%), HLA-B*08 (23%), HLA-B*07 (16%), and HLA-B*44 and B*51 (each 18%) (Appendix A).

### 2.3. HLA-B Alleles in Relation to Clinical Subtypes of SpA

The cohort of 263 HLA-B*27-negative patients comprised 65 individuals with ankylosing spondylitis, 41 with non-radiographic axial SpA, 34 with psoriatic arthritis (including one case of PsA sine psoriasis), 5 with reactive arthritis, and 116 with undifferentiated SpA. Two additional patients fulfilled the criteria for enteropathic arthritis and juvenile-onset SpA, respectively; due to their limited number and the potentially distinct clinical course and underlying pathophysiological mechanisms characteristic of these forms, they were excluded from the subtype-specific allele analysis.

In the subgroup of patients with reactive arthritis, HLA-B*37 and HLA-B*41 appeared disproportionately represented. Three of the five individuals with ReA carried one of these two alleles—an observation suggestive of a potential phenotypic clustering, despite the small sample size (Table 2). In non-radiographic axial SpA, HLA-B*52 was found in 12.2% of cases (5/41), while in undifferentiated SpA, it appeared in only 1 patient out of 116 (*p* = 0.0437; Table 2). Although the overall numbers remain limited, these patterns point toward allele-specific enrichment in distinct SpA subtypes.

When stratified by dominant clinical presentation—axial (*n* = 68), peripheral (*n* = 56), or mixed (*n* = 139)—additional patterns emerged. HLA-B*54, though rare in the general population, was observed exclusively in axial SpA (3/68; *p* = 0.003) and was not detected in patients with peripheral or mixed forms (Table 3). Conversely, HLA-B*35 demonstrated a broad distribution across all clinical subgroups: 32.4% in axial, 28.8% in mixed, and 23.2% in peripheral SpA; however, without statistically significant differences (*p* = 0.5305; Table 3). Other alleles, including HLA-B*07, B*08, and B*18, also showed no distinct phenotype-specific clustering.

### 2.4. HLA-B Allele Distribution and Clinical Phenotype Correlations

To further elucidate the role of non-HLA-B*27 alleles in phenotypic expression, we analyzed allele distribution across clinical presentations, including joint involvement, extra-articular features, and radiographic and laboratory findings, as detailed in Appendix A.

HLA-B*35 and HLA-B*18 remained the most frequently encountered alleles across the cohort, and were consistently overrepresented in patients with axial, peripheral, and systemic features. HLA-B*35 was present in 29.2% of patients with inflammatory low back pain, 21.6% with reduced lumbar mobility, 23.3% with radiographic sacroiliitis, 28.9% with arthritis, 24.4% with enthesitis, and 34.6% with elevated CRP/ESR levels. HLA-B*18 demonstrated similar distribution: 23.2% in inflammatory low back pain, 25.6% in sacroiliitis, 25.2% in arthritis, and 24.4% in enthesitis. Although these patterns did not reach statistical significance (all *p* > 0.05), their repeated co-occurrence supports a potential immunogenetic contribution to extended inflammatory phenotypes.

Peripheral disease features were more frequently associated with HLA-B*07 and HLA-B*08, found in 21.5% and 10.3% of patients with peripheral arthritis, and in 6.7% and 20.0% of those with dactylitis, respectively. HLA-B*07 was also present in 21.9% of patients with enthesitis.

Regarding extra-articular manifestations, HLA-B*35 (33.3%) and HLA-B*18 (18.2%) were the most frequent alleles among patients with cutaneous psoriasis, while HLA-B*08, B*13, and B*44 were each present in 21.2%. In uveitis, HLA-B*18 and B*35 dominated again (28.0% and 24.0%, respectively), followed by B*08 and B*51 (24.0% each). In patients with inflammatory bowel disease, B*08, B*35, and B*51 were each found in 28.6%.

Although most findings did not meet the *p* < 0.05 threshold in Fisher’s test (Appendix A), notable associations included HLA-B*15 with reduced lumbar mobility (*p* = 0.033), B*37 with dactylitis (*p* = 0.017), B*50 with arthritis (*p* = 0.012), and B*57 with psoriasis (*p* = 0.017). Logistic regression confirmed significant associations between cutaneous psoriasis and both HLA-B*13 (OR = 11.29, 95% CI: 1.39–91.75, *p* = 0.023) and HLA-B*57 (OR = 26.01, 95% CI: 2.25–300.58, *p* = 0.009).

Beyond disease phenotype, we also explored potential sex-related immunogenetic differences. While the overall distribution of HLA-B alleles was largely comparable between sexes, select variants exhibited distinct sex-related patterns. HLA-B*41 was disproportionately represented in male patients (8.1%) compared to females (1.9%); however, this also reflects its association with peripheral or mixed forms of SpA, including the ReA and PsA subtypes. This trend was reinforced by logistic regression analysis (OR = 0.20; 95% CI: 0.041–0.961; *p* = 0.044), though the lower confidence bound warrants cautious interpretation. HLA-B*49 followed a similar pattern, being observed in eight male subjects versus three female subjects, possibly reflecting its association with nr-axSpA and USpA subtypes (Appendix A).

### 2.5. Comparative HLA-B Allele Frequencies Between Patients and Controls

To evaluate associations between specific HLA-B alleles and spondyloarthritis susceptibility, allele frequencies in patients were compared with those in two independent HLA-B*27-negative control groups.

The first control group (*n* = 335 subjects) consisted of apparently healthy individuals genotyped using low-resolution typing methods such as HLA typing by sequence-specific primers (SSPs), as previously described in our published study on HLA allele distribution within the Romanian population [8]. All individuals were confirmed to be HLA-B*27-negative and derived from the Bone Marrow Volunteer Donors National Program (January 2011–October 2014). This group served as a previously characterized, ethnically matched comparator cohort.

The second control group (*n* = 1705 subjects) was newly generated for this study and comprised HLA-B*27-negative healthy individuals from the same donor program, genotyped by next-generation sequencing (NGS). This dataset has not been previously reported or analyzed in relation to spondyloarthritis. The large sample size and increased genotyping precision offered enhanced statistical power and improved allelic resolution, allowing for more confident association testing.

In the low-resolution HLA-SSP typing group, HLA-B*47 was significantly more frequent in patients (4.56%) than controls (1.19%; *p* = 0.0189; RR = 3.82, 95% CI: 1.25–11.72; Table 4). HLA-B*53, absent from controls but present in 1.52% of patients (*p* = 0.0369), also emerged as a potential signal, though the absolute frequency was low and the size of the control group was relatively small.

In the NGS genotyping dataset, the association with HLA-B*47 was confirmed (4.56% vs. 1.23%; *p* = 0.0007; RR = 3.71, 95% CI: 1.84–7.44; Table 5). Moreover, HLA-B*54 was notably more frequent among patients than controls (0.76% vs. 0.06%; *p* = 0.0085; RR = 19.45, 95% CI: 2.03–186.4), despite its rarity. In contrast, HLA-B*40 was significantly underrepresented among patients (6.46% vs. 10.85%; *p* = 0.0287; RR = 0.60, 95% CI: 0.37–0.96), suggesting a potential protective association. Furthermore, common alleles such as HLA-B*18 and B*35 showed no significant differences between patients and controls, indicating no apparent role in disease susceptibility despite their high intra-cohort frequencies.

These findings remained consistent when both control groups were pooled into a combined reference population (*n* = 2040 subjects; Table 6), reinforcing the strength and reproducibility of the observed associations.

It is important to mention that HLA associations with spondyloarthritis demonstrate significant population-specific variability, as detailed in Table 7. For example, HLA-B*07 and HLA-B*15 have been identified as risk alleles in North African, Latin American, and South Asian cohorts, while showing neutral or even protective roles in other populations such as Europeans or Han Chinese [9,10,11,12,13,14]. These contrasting patterns underscore the complexity of HLA–disease relationships and highlight the importance of considering genetic background, environmental factors, and population history when interpreting susceptibility signals.

### 2.6. Comparative HLA-B Genotypes and Hardy–Weinberg Disequilibrium Between Patients and Controls

Next, we examined and compared the frequencies of HLA-B genotypes between HLA-B27-negative SpA patients and control subjects. The top four most frequent HLA-B genotypes observed for SpA were B*08-B*18, B*07-B*35, B*13-B*35, and B*35-B*44, while for control subjects were B*08-B*35, B*18-B*35, B*35-B*44, and B*35-B*51, closely followed by B*44-B*51. Applying the Hardy–Weinberg equilibrium (HWE) test, HLA-B genotypes were in disequilibrium for both controls and SpA cases (*p* < 0.0001, Table 8). The HWE analysis confirmed a disequilibrium related to HLA-B*08-B*18, B*13-B*35, and B*15-B*35 in SpA patients for a disequilibrium (D) coefficient value above 2. Interestingly, a set of 10 HLA-B genotypes were only observed within the SpA group (Table 9).

When analyzing the distribution of HLA-B genotypes in SpA patients stratified by axial, mixed, or peripheral forms of the disease, we identified a set of HLA-B genotypes present in at least two of the mentioned subgroups. The top four most frequent shared genotypes were HLA-B*08-B*18, B*15-B*35, B*07-B*35, and B*13-B*35. Conversely, certain genotypes appeared specific to particular SpA forms, such as B*18-B*49 for the axial group, and B*13-B*18, B*35 homozygosity, or B*35-B*53 for the mixed group (Table 10).

We next explored the genotype–phenotype correlations by stratifying patients based on radiographic damage or inflammatory markers (elevated CRP or ESR). Thus, the HLA-B genotypes and allele frequencies were computed in patients with radiologically confirmed sacroiliitis grade ≥ 2. The Hardy–Weinberg equilibrium test indicated significant disequilibrium (*p* < 0.0001), and several HLA-B allele combinations were indeed more frequent in this group, with a D coefficient value above 2. These included B*08-B*18, B*40-B*44, B*07-B*44, and B*07-B*18 (Table 11 and Appendix A). Additionally, certain genotypes were only present in this subgroup and involved the HLA-B*40 allele, such as B*40-B*44, B*40-B*50, and B*07-B*40. When stratifying the patients based on inflammatory markers, several genotypes were more frequent within the group with elevated CRP and ESR (B*08-B*18, B*40-B*44, B*07-B*44, B*35-B*44), while other were more frequent in patients with no paraclinical signs of inflammation (e.g., B*13-B*35, Table 12 and Appendix A).

A heatmap displaying the frequency of HLA-B genotypes and individual alleles enables the visualization of the presented differences between the SpA groups with confirmed radiological damage (Figure 1A) or elevated inflammatory markers (Figure 1B). Thus, we observed a lower frequency of HLA-B*35 among patients with radiologically confirmed sacroiliitis grade ≥ 2, whereas its frequency was elevated in the cohort characterized by inflammatory biomarkers (elevated CRP or ESR).

The interpretation of Hardy–Weinberg disequilibrium (HWD) data for HLA-B genotypes in HLA-B*27-negative spondyloarthritis patients requires caution due to several potential confounding factors. Firstly, similar to linkage disequilibrium, the relatively small sample size (*n* = 263) may also result in unstable disequilibrium (D) estimates, particularly for low-frequency alleles. Moreover, unequal representation of patients based on disease severity, the presence of paraclinical markers (such as inflammation or radiographic changes), or specific clinical features may introduce selection bias. Another important consideration is the underlying population structure of the study cohort, as the presence of distinct ethnic subgroups could influence genotype distribution. However, information on ethnic background was not consistently available for all participants, limiting the ability to control for this potential confounder.

### 2.7. Comparative Peptide-Binding of Risk HLA-B Alleles Versus HLA-B*27

HLA polymorphism is critically important for determining peptide-binding specificity. In HLA-I molecules, the peptide-binding groove is formed by the α1 and α2 domains, which together create a cleft composed of six structurally distinct pockets termed A–F. These pockets accommodate the side chains of peptide residues, with pockets B and F typically engaging the primary anchor residues as the position 2 (P2) of the peptide and the C-terminal residue, respectively. The specific amino acid composition of each pocket, shaped by allelic variation, defines the biochemical environment and determines which peptide side chains can be accommodated. As a result, different HLA alleles bind distinct peptide repertoires, and their functional similarity is defined by the sets of peptides they present and the binding affinity for those peptides [37,38].

Considering this, we next asked whether the identified risk alleles differ from the well-characterized HLA-B*27 allele associated with SpA. To investigate this, we compared the IC_50_ values of HLA-B alleles for binding two distinct sets of peptides, previously published in *Molecular and Cellular Proteomics* (by Laura Cobos-Figueroa et al., 2025 [39]) and *Angewandte Chemie International Edition* in English (by Liangliang Sun et al., 2014 [40]) journals.

To generate the IC_50_ values for each combination of selected HLA-B alleles and peptides, we used the Immune Epitope Database (IEDB) Analysis Resource (http://tools.iedb.org/, accessed on 1 July 2025). In our analysis, we considered values below 50 nM as high-affinity binding, values between 50 and 500 nM as intermediate-affinity binding, and values above 500 up to 5000 nM as low-affinity binding. First, we trimmed the list of peptides and included only those with an IC_50_ value below 5000 nM for HLA-B*27. Next, we generated a heatmap to assess correlations between HLA-B*27 and selected risk alleles. The analysis of the recently published set of peptides in *Molecular and Cellular Proteomics* [39] (Appendix A) revealed notable similarities between HLA-B*18 and HLA-B*57, as well as between HLA-B*51 and HLA-B*54, while overall showing a low correlation between the selected risk alleles and HLA-B*27 (Figure 2A). Interestingly, a negative correlation was observed between B*27 and B*51 (correlation coefficient = −0.45). Additionally, two other moderately correlated clusters involving HLA-B*35 emerged: one comprising B*35, B*07, and B*08; and another including B*35, B*18, and B*40. By using the broader set of peptides [40] (Appendix A), the previous correlation clusters expanded, including also B*54 and B*57 (Figure 2B).

We then examined the set of peptides for which at least one HLA-B allele showed high binding affinity and visualized the results using a heatmap. While many of the risk alleles demonstrated high affinity for overlapping peptide sets, the repertoire bound by HLA-B*27 appeared notably distinct (Figure 2C). To explore potential relationships among the SpA risk alleles, we plotted the 9-mer peptides bound with high affinity by at least two of them. This analysis revealed several patterns of similarity, such as those among B*07, B*08, B*35, B*51, and B*54; between B*35 with either B*07, B*08, B*18, or B*54; between B*18 with either B*40 or B*54, and reduced similarities between B*27 with either B*08, B*18, or B*35.

Considering the marked differences in the peptide-binding patterns of HLA-B*27 compared to the other risk alleles identified in this study, it is highly likely that distinct pathogenic mechanisms are involved in HLA-B*27-associated versus B*27-negative spondyloarthritis. This suggests that HLA-B*27-negative SpA may represent a separate clinical and immunological entity, warranting a deeper and more specific understanding of its underlying pathogenesis.

## 3. Discussions

### 3.1. Cross-Reactivity and the Diagnostic Hypothesis

This study centered on a cohort from the historical region of Moldova in Northeastern Romania—an Eastern European population historically underrepresented in immunogenetic studies of spondyloarthritis, thereby addressing a critical gap in the existing literature. Building on the diagnostic insights of Cianga et al. (2011), who demonstrated flow cytometric cross-reactivity between HLA-B27 antibodies and structurally related HLA-B alleles such as B07, B18, and B35 [7], we sought to determine which of the HLA-B alleles shows a genuine immunogenetic association with disease.

Our analysis did not reveal significant associations for most of the initially presumed cross-reactive antigens. Among them, HLA-B*07 was present in 14.07% of SpA patients compared to 11.03% of controls (*p* = 0.1462; RR = 1.28 [0.92–1.77]). Though not statistically significant, it showed increased prevalence in patients with enthesitis (21.95%) and arthritis (10.28%) (Appendix A), suggesting a potential, albeit unconfirmed, role in peripheral disease. However, considering the high baseline frequency of HLA-B*07 in European populations, including Romania (~6.4%) [8], this pattern may equally reflect population background rather than a disease-specific association, warranting caution in interpretation.

HLA-B*18, similarly flagged by Cianga et al. for its moderate cross-reactivity [7], was found in 22.43% of patients and 19.41% of controls (*p* = 0.2460; RR = 1.16 [0.91–1.48]). Although this represents a modest numerical increase, it falls within the expected margin of population variation and does not support a pathogenic role. However, its recurring presence across axial and peripheral clinical features, including inflammatory back pain (23.24%), reduced spinal mobility (20.67%), radiographic sacroiliitis (25.56%), arthritis (25.23%), and enthesitis (24.39%, and systemic inflammation (24.36%) (Appendix A), suggests a possible role in shaping an inflammatory phenotype.

Likewise, HLA-B*35 was found in 28.52% of SpA patients versus 26.04% of controls (*p* = 0.4085; RR = 1.10 [0.89–1.35]), and was particularly frequent in patients with inflammatory back pain (29.19%), radiographic sacroiliitis (23.33%), arthritis (28.97%), enthesitis (24.39%), and psoriasis (33.33%), as well as those with elevated systemic inflammatory markers (34.62%) (Appendix A). These findings, although not statistically significant, may suggest a role in modulating disease severity or tissue involvement.

Other molecules previously implicated in moderate cross-reactivity—namely HLA-B37, B38, and B39—were also examined in our cohort. None demonstrated a significant difference in frequency between patients and controls. HLA-B*37 was found in 1.52% of patients and 1.88% of HLA-NGS genotyped controls (*p* = 1.000), HLA-B*38 in 9.89% vs. 8.45% (*p* = 0.4114), and HLA-B*39 in 4.18% vs. 5.16% (*p* = 0.6486). However, HLA-B*37, although overall rare, was detected in 13.33% of patients with dactylitis and in 20% of those with reactive arthritis (Appendix A), underscoring the possibility of subtype-specific associations not captured in aggregate analyses. Conversely, the descriptive analysis suggested the potential involvement of additional actors, though these were not significantly enriched within the overall cohort. Specifically, HLA-B*08 and HLA-B*51 were each observed in 24% of patients with uveitis and in 28.57% of those with inflammatory bowel disease. Similarly, HLA-B*13 and HLA-B*57 showed associations with cutaneous psoriasis, with odds ratios of 11.29 (*p* = 0.023; 95% CI: 1.39–91.75) and 26.01 (*p* = 0.009; 95% CI: 2.25–300.58), respectively. Nevertheless, due to limited statistical power and wide confidence intervals, these findings should be interpreted with caution.

### 3.2. HLA-B*47, B*54, and B*40: Novel Associations Beyond B27

#### 3.2.1. HLA-B47: Association with SpA and Structural Relatedness to HLA-B27

In contrast to previously hypothesized cross-reactive molecules, our study identified significant associations with other, less-anticipated HLA-B variants. Notably, HLA-B*47 emerged as a strong candidate risk allele, being present in 4.56% of SpA patients compared to 1.23% of controls (*p* = 0.0007; RR = 3.705 [1.844–7.441]). HLA-B*47 has an average national prevalence of 0.6% in Romania, with a marked regional gradient—from 0.1% in Moldova to 1.0% in Wallachia [8]. Given that all patients in our cohort originated from Moldova, its significant enrichment among SpA patients, despite extreme local rarity, suggests a potential role as a rare, high-penetrance risk allele.

This association invites further investigation into the immunopathogenic mechanisms by which HLA-B47 may contribute to disease. HLA-B47 is classified within the B27 supertype, a cluster of HLA class I molecules sharing conserved structural features, particularly at the level of the peptide-binding groove [38].

Moreover, experimental monoclonal antibody studies have provided indirect evidence of structural similarity between HLA-B47 and HLA-B27. One such antibody, TrBH12, was derived from Epstein–Barr virus-transformed peripheral blood mononuclear cells of a multiparous donor who had developed HLA-B27-specific alloantibodies. Although designed to recognize HLA-B27, TrBH12 was shown to cross-react not only with B27-positive cells, but also with lymphoblastoid cell lines expressing HLA-B37 and Bw47, a serologic equivalent of HLA-B47. This reactivity pattern suggests the presence of a shared conformational epitope among these alleles, reinforcing its plausibility as a candidate allele involved in spondyloarthritis pathogenesis [41]. While the peptide-binding specificity of HLA-B47 is not yet comprehensively characterized, its inclusion within the B27 supertype suggests an overlap in peptide-binding preferences, with potential implications for CD8+ T cell-mediated antigen presentation. This raises the possibility that HLA-B47, similar to HLA-B27, may present arthritogenic peptides capable of initiating or sustaining aberrant T cell responses—an established hallmark of spondyloarthritis pathogenesis. Although HLA-B47 has been predominantly studied in the context of congenital adrenal hyperplasia, particularly in association with 21-hydroxylase deficiency [42], its immunogenetic profile and rare but consistent enrichment in SpA patients positions it as a candidate allele worthy of further exploration in autoimmune contexts.

#### 3.2.2. HLA-B*54: A Rare Allele Potentially Involved in axSpA Susceptibility

Secondly, HLA-B*54, an allele with extremely low prevalence both in Romania and in broader European populations [8], demonstrated a similarly disproportionate representation in our SpA cohort, particularly among individuals with axial disease. Although its national frequency averages 0.1%, regional data report up to 0.6% in Banat. In contrast, HLA-B54 is virtually absent in Moldova, the geographic origin of our cohort. Its complete absence from controls and clustering in axSpA cases suggests a localized, possibly founder effect-driven contribution to disease susceptibility.

From an immunological standpoint, HLA-B54 belongs to cross-reactive group (CREG) 7, which encompasses molecules such as HLA-B27 and HLA-B40, known to share overlapping public epitopes and structural motifs [43]. Structural modeling studies suggest that HLA-B54 shares critical amino acid substitutions with HLA-B27 within the α1 helix of the peptide-binding groove and exhibits partially conserved residues within the α2 domain—regions critical for peptide affinity and interaction with T cell receptors. These shared features raise the possibility that HLA-B54 may present peptide repertoires that partially overlap with those of HLA-B27, including peptides with immunogenic or arthritogenic potential. Furthermore, specific polymorphisms in the α2 domain of HLA-B54 have been implicated in T cell cross-reactivity, suggesting that its structural configuration may facilitate autoreactive immune responses in genetically susceptible individuals [43].

Additional support for the immunomodulatory role of HLA-B54 derives from its established association with diffuse pan-bronchiolitis (DPB), a chronic inflammatory pulmonary disorder characterized by marked CD8+ T cell infiltration, elevated levels of IL-8 and macrophage inflammatory protein-1, and persistent neutrophilic inflammation [44,45]. Of particular interest is the chronic colonization with *Pseudomonas aeruginosa*, detected in 22% of DPB patients at diagnosis and rising to over 60% after four years [46]—a scenario that mirrors chronic mucosal immune activation and sustained antigenic pressure, both of which are implicated in SpA pathogenesis. While DPB is a respiratory condition and SpA targets the axial skeleton and entheses, the shared features of neutrophil-rich inflammation, CD8+ T cell involvement, and environmental microbial triggers suggest a possible convergence of immunological mechanisms in HLA-B54–mediated pathology.

Furthermore, the association of HLA-B54 with reactive arthritis following influenza vaccination in a 79-year-old Japanese patient [47] adds additional evidence that this allele may predispose to aberrant immune activation in the context of non-infectious antigens. Although this is an isolated case report, it supports the hypothesis that HLA-B54 can facilitate pathological T cell responses under specific antigenic stimuli, further reinforcing its plausibility as an SpA risk allele.

Still, these allele-specific signals must be interpreted within the broader genomic architecture of the MHC, where genes are not inherited independently but as components of extended haplotypes shaped by evolutionary selection and recombination constraints. Within this context, the observed associations with HLA-B*47 and B*54 may not solely reflect the direct pathogenic role of these alleles but could also suggest broader linkage disequilibrium blocks encompassing adjacent immunoregulatory genes. The MHC region harbors numerous loci involved in antigen processing (e.g., TAP), cytokine signaling (e.g., TNF), and T cell activation (e.g., MICA) [48,49]—any of which may contribute to SpA susceptibility when co-inherited with certain HLA-B variants. In this light, HLA-B*47 and -B*54 may act less as isolated genetic risks and more as sentinel alleles within structurally and functionally integrated haplotypes that collectively shape immunological thresholds for inflammation.

#### 3.2.3. HLA-B*40: Protective Association in the Studied Romanian Cohort

In contrast to these rare, putatively high-impact associations, HLA-B*40 exhibited a statistically significant inverse correlation in our cohort, suggesting a potential protective role against SpA. Specifically, HLA-B*40 was present in 6.46% of patients versus 10.85% of controls (*p* = 0.0287; RR = 0.596). Notably, HLA-B*40 is moderately prevalent across Romania, with an average frequency of 4.74% and minimal regional variation [8], suggesting that its protective effect may operate across a broader segment of the population. Unlike B*47 and B*54, which appear to confer phenotype-specific risk within narrowly defined genetic backgrounds, HLA-B*40 may exert a broader protective influence, consistent with its relatively uniform population distribution. This effect may reflect underlying differences in peptide-binding properties or regulatory interactions within MHC genotypes, although the precise immunological mechanisms remain to be clarified.

#### 3.2.4. HLA-B*53: A Modest Frequency Signal of Potential Relevance

A modest signal was also noted for HLA-B*53, which was present in 1.52% of SpA patients and absent from the HLA-SSP typing control group (*p* = 0.0369). In the HLA-NGS typing comparison, B*53 remained more frequent in patients (1.52%) than in controls (0.59%), although the association did not reach statistical significance (*p* = 0.1292; RR = 2.357). The average reported frequency of B*53 in Romania is approximately 0.5%, with minimal regional variation [8], and the distribution observed in our Moldovan cohort aligns with this background level. While the low-resolution signal may reflect random fluctuation or cohort-specific noise, it is also possible that the rarity of the allele limited the statistical power to detect a meaningful association in the larger control set.

### 3.3. Population-Specific HLA-B Patterns in SpA

Taken together, our findings indicate that while cross-reactivity provided a logical entry point, structural similarity alone is not a reliable predictor of genetic risk. Several molecules previously implicated in flow cytometric interference with HLA-B27 showed no significant association with SpA in our cohort, underscoring the need to differentiate diagnostic artifact from true immunogenetic relevance. These results support the need to reframe SpA genetics beyond a B*27-centric lens, setting the stage for a more integrated understanding of disease heritability across both HLA and non-HLA loci.

To contextualize our findings, we examined previously reported associations between non-HLA-B*27 HLA-B alleles and spondyloarthritis across diverse populations (Table 7*)*. The landscape of non-HLA-B*27 associations in spondyloarthritis is notably shaped by population-specific genetic architectures. Certain HLA-B alleles display robust, replicated associations within defined ethnic and geographic contexts, but fail to achieve significance elsewhere, highlighting the limitations of extrapolating genetic risk across populations. For instance, HLA-B*07 has emerged as a strong risk factor in French axSpA cohorts and in Tunisian, Brazilian, and Western Indian populations with USpA, and has also been linked to reactive arthritis and idiopathic sacroiliitis in American patients [9,10,11,12]. Conversely, the same allele has demonstrated protective associations with AS in European, Han Chinese, and African American populations [13,14], underscoring the bidirectional nature of HLA–disease relationships and the necessity of context-aware interpretation. Meanwhile, HLA-B*15 has demonstrated a particularly compelling signal in North African and Latin American USpA cohorts [10,21,22]. Similarly, HLA-B*38 has been associated with AS in both Southern European and Anglo-American cohorts, and also appears in PsA populations from Argentina and Canada [14,27,28,29]. HLA-B*35 also exhibits phenotype-specific associations, particularly in Croatians with USpA, where it correlates with sacroiliitis, peripheral arthritis, and systemic inflammation at clinically significant levels [25]. These findings illustrate that genetic susceptibility to SpA is not uniformly distributed, but instead reflects complex historical, demographic, and evolutionary forces that shape allele frequency, linkage disequilibrium or HWD, and immunological function at the population level.

Within this context, our study addresses a critical gap in Eastern European immunogenetics and reinforces the need to interpret SpA risk within its population-specific context, rather than extrapolating from external association signals. Our findings expand the understanding of HLA-B involvement in B27-negative SpA and underscore the value of regionally focused studies, highlighting HLA-B*47 and B*54 as candidate susceptibility markers. By combining deep clinical phenotyping with HLA genotyping, this study offers novel insights and a replicable framework for future studies in underrepresented populations. Interestingly, in our previous study on HLA associations in myasthenia gravis, another autoimmune disease, we also identified HLA-B*08, B*44, B*47, and B*57 as risk alleles [50]. This finding may suggest shared immunogenetic mechanisms across different autoimmune diseases and inflammatory processes or reflect population-specific genetic patterns characteristic of the Northeastern region of Romania.

The analysis of HLA-B genotypes reveals the predominance of several combinations within the HLA-B*27-negative SpA cohort, including B*08-B*18, B*07-B*35, B*13-B*35, and B*35-B*44. These simultaneous associations of two HLA-B risk alleles may result in a synergistic effect, contributing to a stronger genetic determinism in disease susceptibility than that conferred by each individual allele alone. While Hardy–Weinberg equilibrium testing confirmed significant deviations for these genotypes, suggesting true non-random allele associations, multiple potential confounders may still influence the observed disequilibrium patterns, similar to linkage disequilibrium. Tools such as PHASE, along with reference data from the 1000 Genomes Project, highlight several sources of bias that should be considered when interpreting disequilibrium estimates [51,52]. These include the relatively small sample size and the presence of rare alleles, population stratification, as well as potential genotyping errors. Although the HWE results support the existence of HW disequilibrium, these additional factors may impact the stability, accuracy, and generalizability of the findings and should be carefully considered when drawing conclusions.

One should also consider the relevance of investigating linkage disequilibrium (LD) involving other genes located within the HLA class I region, which are also known to influence the evolution and clinical expression of spondyloarthritis. Among these, particular attention has been given to the MICA genes, due to their immunomodulatory role and genomic proximity to HLA-B. Several studies have reported associations between specific MICA alleles and susceptibility to spondyloarthritis (SpA), particularly MICA*007, MICA*008, and MICA*009, which have been implicated in increased risk or modulation of disease severity [53,54]. These associations may reflect both functional immunological roles of MICA (e.g., via NKG2D receptor activation), and linkage disequilibrium (LD) with nearby HLA-B alleles. For instance, the population data retrieved from www.allelefrequencies.net (accessed on 1 July 2025) highlighted distinct MICA–HLA-B allele associations observed in two distinct United States Caucasian cohorts, as detailed in Table 13. Remarkably, MICA*008 was more frequently associated with the HLA-B risk alleles B*07, B*08, B*40, and B*44 reported herein, and MICA*009 was linked with B*35, while MICA*007 with B*27.

These patterns may reflect underlying linkage disequilibrium structures within the HLA class I region, suggesting that certain extended haplotypes, involving both MICA and HLA-B alleles, could also contribute to genetic susceptibility or modulation of disease phenotype in spondyloarthritis. Future studies should thus consider such MICA–HLA-B associations in the context of both genetic risk assessment and functional immunogenetic analysis in spondyloarthritis.

### 3.4. Study Limitations and Further Directions

Several limitations of this study should be acknowledged. The sample size, though substantial for a single-center cohort from the historical region of Moldova, limits the statistical power of subgroup analyses, particularly for rare alleles such as HLA-B*54. While the associations observed for HLA-B*47 and B*54 reached statistical significance, their low absolute frequencies warrant cautious interpretation and demand replication in larger, independent cohorts. Importantly, this study focused on allele-level associations and did not explore the functional consequences of the implicated variants. Further research is needed to elucidate the immunobiological roles of these alleles, including their impact on antigen presentation, peptide repertoire, T cell activation, and interactions with endoplasmic reticulum aminopeptidases—mechanisms central to SpA pathogenesis.

In addition, the absence of standardized assessments of disease activity and treatment response limits conclusions regarding the clinical expression or therapeutic implications of specific HLA-B variants. Integrating immunogenetic data with validated activity indices (e.g., BASDAI, ASDAS), treatment outcomes, and longitudinal phenotyping would allow a more complete understanding of how genetic variation shapes disease course and response to therapy.

Future research should also include haplotype-based approaches to better reflect the block-like inheritance patterns of the MHC and associated linkage disequilibrium. Within this highly linked region, HLA-B alleles likely operate not in isolation but as components of broader configurations that include other immunoregulatory HLA and non-HLA genes. Capturing these inherited blocks may help clarify the combinatorial effects of multiple variants, some subtle yet synergistic, on SpA susceptibility, phenotype expression, and therapeutic response.

## 4. Materials and Methods

### 4.1. Study Design and Population

This study included HLA-B*27-negative patients with suspected spondyloarthritis evaluated at the Clinical Rehabilitation Hospital of Iași and genotyped at the Immunology Department of “Grigore T. Popa” University of Medicine and Pharmacy. Patients were identified from existing institutional records or newly referred for clinical assessment and HLA-B typing. A total of 263 individuals meeting ASAS classification criteria for axial or peripheral SpA were included in the final analysis after exclusion of differential diagnoses.

### 4.2. Inclusion and Exclusion Criteria

Eligible patients were aged ≥16 years and presented with clinical features suggestive of SpA, including inflammatory back pain (≥3 months), peripheral arthritis, enthesitis, or dactylitis. Inclusion required fulfillment of ASAS classification criteria and complete medical records, including clinical, imaging, and laboratory data.

Exclusion criteria were HLA-B*27 positivity, alternative autoimmune or degenerative conditions (e.g., rheumatoid arthritis, lupus, osteoarthritis), or a history of malignancy or systemic immune disorders.

### 4.3. Genotyping Protocol

HLA-B low-resolution genotyping for control group 1 was conducted using Polymerase Chain Reaction with sequence-specific primers (SSPs), employing commercially available kits from BAG (BAG Health Care GmbH, Lich, Germany) and Olerup (Olerup SSP AB, Stockholm, Sweden), as described previously [50]. The HLA genotyping for control group 2 was performed through high-resolution next-generation sequencing (NGS) technology, utilizing the NGSgo-MX6–1 kit (GenDx, Utrecht, The Netherlands) on an iSeq Illumina platform (Illumina, San Diego, CA, USA). The analysis was carried out using GenDx NGSengine v2.22 (GenDx, Utrecht, The Netherlands). However, the results for control group 2 were also reported at a two-digit (low-resolution) level.

### 4.4. Statistical Analysis

Descriptive analyses were performed to quantify HLA-B allele frequencies across the full SpA cohort and stratified by sex, diagnostic subtype, axial versus peripheral disease patterns, and extra-articular manifestations. Associations between individual alleles and clinical features, such as inflammatory back pain, sacroiliitis, psoriasis, uveitis, or elevated inflammatory markers, were evaluated using Fisher’s exact test. Binary logistic regression was employed to assess the association between allele presence (independent variable) and specific clinical traits (dependent variable, coded as present/absent), with results expressed as odds ratios (ORs) and 95% confidence intervals (CIs). Comparative analyses between patients and two HLA-B*27-negative Romanian control groups (*n* = 335 and *n* = 1705 subjects) were conducted using chi-square or Fisher’s exact tests, as appropriate. Relative risks (RRs) with 95% CI were calculated to estimate the strength and direction of association between individual alleles and SpA susceptibility. All statistical analyses were performed using SPSS Statistics v26.0 (IBM Corp., Armonk, NY, USA) and GraphPad Prism v10.0 (GraphPad Software, San Diego, CA, USA). Bootstrap resampling was employed in the multivariate analysis to improve the stability of estimates in subgroups with small sample sizes, such as reactive arthritis (ReA), which included only 5 cases, using SPSS v26.0. The correlation coefficients were computed using GraphPad Prism v10.0 and the corresponding heatmaps were generated in OriginPro v.9.2 (OriginLab Corporation, Northampton, MA, USA). The Hardy–Weinberg equilibrium (HWE) was assessed using the chi-squared test, following the methodology described by Chen et al. [57]. Similar to the linkage disequilibrium (LD), the disequilibrium coefficient (D) was computed as the difference between the observed frequency of a given genotype or haplotype (P_XY_) and the expected frequency under independence (P_X_ × P_Y_). The IC_50_ values for assessing the binding affinity between HLA-B alleles and various peptides were computed using the Immune Epitope Database (IEDB) Analysis Resource (http://tools.iedb.org/, accessed on 1 July 2025). A *p*-value of less than 0.05 was considered statistically significant.

### 4.5. Ethical Considerations

The study was conducted in accordance with the Declaration of Helsinki and approved by the Ethics Committees of the Clinical Rehabilitation Hospital and the “Grigore T. Popa” University of Medicine and Pharmacy, Iași (IRB number: 102/12.07.2021).

Written informed consent was obtained from all prospectively enrolled participants, and retrospective data were handled under ethical exemption. For the retrospective cohort, data were accessed under ethical exemption and in compliance with national data protection regulations. As these patients were no longer in follow-up, recontact was not feasible. At the time of admission, patients (or their guardians, for minors) had provided general consent allowing the use of anonymized data for research purposes. All data analyzed were fully anonymized prior to inclusion in this study.

## 5. Conclusions

Our study presents the first detailed immunogenetic characterization of an HLA-B*27-negative spondyloarthritis cohort from Romania. Through comparative analysis with ethnically matched controls and phenotype-based stratification, we identified both allele-specific distributions and possible genotype–phenotype correlations.

Notably, this is the first study to report an enrichment of HLA-B*47 and HLA-B*54—the latter observed exclusively in patients with axial SpA—alongside a relative underrepresentation of HLA-B*40 in HLA-B*27-negative SpA patients, suggesting potential allele-specific contributions to disease susceptibility and protection. These findings complement the broader intra-cohort analysis, where HLA-B*35 and HLA-B*18 emerged as the most frequent alleles across clinical subtypes, and further descriptive patterns revealed the clustering of HLA-B*13 and HLA-B*57 in psoriatic cases. Additionally, the most frequently observed HLA-B genotype combinations among SpA patients were B*08-B*18, B*13-B*35, and B*35-B*51. The B*08-B*18 genotype was particularly associated with more advanced structural damage, as reflected by radiographic sacroiliitis grade ≥ 2, whereas B*35-B*51 was more prevalent among patients exhibiting systemic inflammation, indicated by elevated CRP or ESR levels. Additionally, the distinct peptide-binding repertoires observed within the risk allele cluster (HLA-B*08, B*18, B*35, B*40, and B*54) implicate distinct immunopathogenic mechanisms in spondyloarthritis. These findings may support the concept that HLA-B*27-negative spondyloarthritis represents a separate clinical and immunological entity, reinforcing the need for a more comprehensive understanding of the underlying inflammatory pathways, which could lead to personalized targeted treatments specific to these pathogenic processes.

By focusing on an Eastern European population historically underrepresented in immunogenetic research, this work underscores the value of geographically and genetically specific cohorts in refining our understanding of HLA-mediated risk. The results call for a re-evaluation of current diagnostic paradigms that privilege HLA-B*27 at the expense of broader genotyping, particularly in early or seronegative disease presentations.

## Figures and Tables

**Figure 1 ijms-26-07617-f001:**
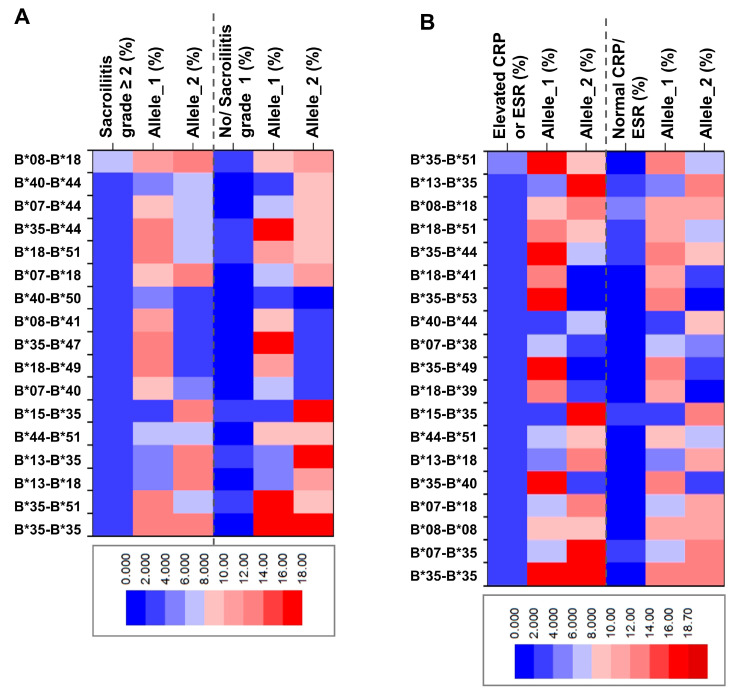
Heatmap of frequency distribution of HLA genotypes and individual alleles among HLA-B*27-negative SpA patients stratified by (**A**) radiological sacroiliitis grade (≥2 or below) and (**B**) inflammation (elevated or normal CRP and ESR).

**Figure 2 ijms-26-07617-f002:**
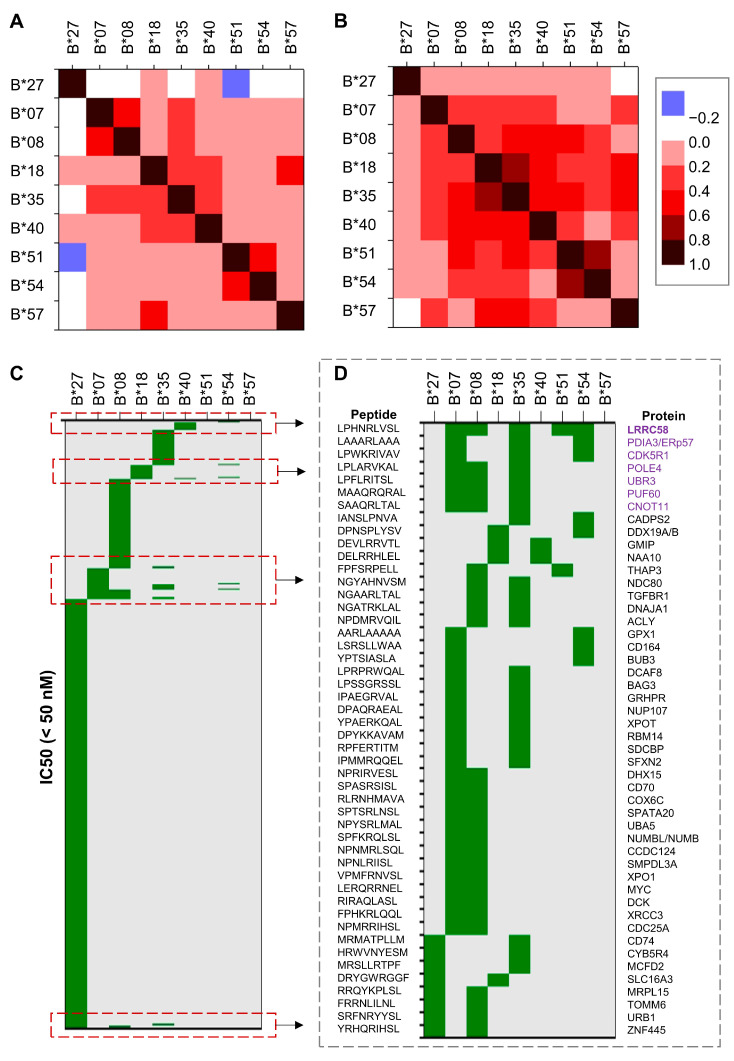
Heatmap of theoretical correlations between distinct HLA-B alleles and distinct sets of peptides. Heatmap of IC_50_ correlation coefficients between distinct HLA-B alleles and (**A**) a set of human peptides known to bind HLA-B molecules, and (**B**) a broader set of peptides. (**C**) Heatmap of peptides with IC_50_ values < 50 nM marked in green (high affinity interaction); (**D**) examples of 9mer peptides binding to distinct HLA-B alleles with IC_50_ below 50 nM. The proteins containing peptides bound by at least 3 distinct HLA-B molecules are highlighted in purple.

**Table 1 ijms-26-07617-t001:** Demographic and phenotypic characteristics of the HLA-B*27-negative spondyloarthritis cohort (*n* = 263).

Category	Clinical Feature	*n* (%)
Musculoskeletal	Inflammatory low back pain	185 (70.3%)
	Peripheral joint involvement	187 (71.1%)
	Lower limb involvement	157 (59.7%)
	Upper limb involvement	118 (44.9%)
	Abnormal spinal mobility (Schober < 5 cm)	208 (79.1%)
	Enthesitis	41 (15.6%)
	Dactylitis	15 (5.7%)
	Heel pain	47 (17.9%)
Extra-articular	Psoriasis	33 (12.5%)
	Uveitis	25 (9.5%)
	Inflammatory bowel disease	7 (2.7%)
Laboratory	Elevated CRP or ESR	78 (29.7%)
Imaging	Sacroiliitis (grade ≥ 2)	90 (34.2%)
	MRI-detectable lesions or grade 1 changes	108 (41.1%)
	Symmetrical sacroiliitis	118 (44.9%)

**Table 2 ijms-26-07617-t002:** Distribution of HLA-B alleles across diagnostic subtypes in HLA-B*27-negative SpA patients.

HLA-B	AS (*n* = 65)	nr-axSpA (*n* = 41)	PsA (*n* = 34)	ReA (*n* = 5)	USpA (*n* = 116)	*p*-Value	Bootstrap *p*-Value
HLA-B*07	10 (15.38%)	9 (21.95%)	6 (17.65%)	0 (0%)	12 (10.34%)	0.3220	0.441
HLA-B*08	12 (18.46%)	7 (17.07%)	7 (20.59%)	1 (20%)	18 (15.52%)	0.9626	0.273
HLA-B*13	7 (10.77%)	3 (7.32%)	6 (17.65%)	0 (0%)	12 (10.34%)	0.5890	0.729
HLA-B*14	4 (6.15%)	2 (4.88%)	**3 (8.82%)** **^†^**	0 (0%)	1 (0.86%)	0.1746	0.305 (**0.040 ^†^**)
HLA-B*15	5 (7.69%)	1 (2.44%)	2 (5.88%)	0 (0%)	11 (9.48%)	0.5979	0.700
HLA-B*18	17 (26.15%)	7 (17.07%)	4 (11.76%)	1 (20%)	30 (25.86%)	0.3807	0.472
HLA-B*35	14 (21.54%)	15 (36.59%)	8 (23.53%)	0 (0%)	38 (32.76%)	0.1805	0.173
HLA-B*37	1 (1.54%)	0 (0%)	0 (0%)	**1 (20%)** **^†^**	2 (1.72%)	**0.0140**	0.437 (**0.045 ^†^**)
HLA-B*38	5 (7.69%)	5 (12.2%)	2 (5.88%)	1 (20%)	13 (11.21%)	0.7368	0.879
HLA-B*39	2 (3.08%)	0 (0%)	2 (5.88%)	0 (0%)	7 (6.03%)	0.4904	0.492
HLA-B*40	6 (9.23%)	1 (2.44%)	3 (8.82%)	1 (20%)	6 (5.17%)	0.4006	0.669
HLA-B*41	2 (3.08%)	2 (4.88%)	2 (5.88%)	2 (40%) ^†^	4 (3.45%)	**0.0045**	0.325 (**0.005 ^†^**)
HLA-B*44	10 (15.38%)	2 (4.88%)	8 (23.53%)	1 (20%)	17 (14.66%)	0.2447	0.116
HLA-B*47	3 (4.62%)	2 (4.88%)	1 (2.94%)	0 (0%)	6 (5.17%)	0.9686	0.986
HLA-B*48	1 (1.54%)	1 (2.44%)	0 (0%)	0 (0%)	1 (0.86%)	0.8741	0.955
HLA-B*49	2 (3.08%)	3 (7.32%)	1 (2.94%)	1 (20%)	4 (3.45%)	0.3335	0.782
HLA-B*50	4 (6.15%)	1 (2.44%)	1 (2.94%)	0 (0%)	3 (2.59%)	0.7324	0.910
HLA-B*51	9 (13.85%)	3 (7.32%)	4 (11.76%)	0 (0%)	22 (18.97%)	0.3273	0.402
HLA-B*52	4 (6.15%)	**5 (12.2%)** **^†^**	2 (5.88%)	0 (0%)	1 (0.86%)	0.0437	0.110 (**0.013 ^†^**)
HLA-B*53	1 (1.54%)	1 (2.44%)	1 (2.94%)	0 (0%)	1 (0.86%)	0.8953	0.978
HLA-B*54	1 (1.54%)	0 (0%)	1 (2.94%)	0 (0%)	1 (0.86%)	0.7968	0.657
HLA-B*55	1 (1.54%)	2 (4.88%)	1 (2.94%)	0 (0%)	2 (1.72%)	0.7848	0.967
HLA-B*56	2 (3.08%)	0 (0%)	0 (0%)	0 (0%)	3 (2.59%)	0.6796	0.701
HLA-B*57	3 (4.62%)	1 (2.44%)	2 (5.88%)	0 (0%)	3 (2.59%)	0.8383	0.948
HLA-B*58	1 (1.54%)	1 (2.44%)	1 (2.94%)	0 (0%)	2 (1.72%)	0.9808	0.997

Note: AS, ankylosing spondylitis; nr-axSpA, non-radiographic axial spondyloarthritis; PsA, psoriatic arthritis; ReA, reactive arthritis; USpA, undifferentiated spondyloarthritis; *n*, number of individuals; D, disequilibrium coefficient; HW, Hardy–Weinberg equilibrium test; bold, cases with *p*-values < 0.05; ^†^, the SpA subtype with *p*-values < 0.05 (multivariate analysis).

**Table 3 ijms-26-07617-t003:** HLA-B allele frequencies in HLA-B*27-negative SpA patients stratified by axial, peripheral, and mixed phenotypes.

HLA-B	Axial (*n* = 68)	Peripheral (*n* = 56)	Mixed (*n* = 139)	*p*-Value
HLA-B*07	11 (16.18%)	7 (12.5%)	19 (13.67%)	0.8260
HLA-B*08	11 (16.18%)	13 (23.21%)	23 (16.55%)	0.4995
HLA-B*13	7 (10.29%)	7 (12.5%)	14 (10.07%)	0.8784
HLA-B*14	3 (4.41%)	3 (5.36%)	4 (2.88%)	0.6825
HLA-B*15	3 (4.41%)	6 (10.71%)	10 (7.19%)	0.4024
HLA-B*18	14 (20.59%)	11 (19.64%)	34 (24.46%)	0.7005
HLA-B*35	22 (32.35%)	13 (23.21%)	40 (28.78%)	0.5305
HLA-B*37	1 (1.47%)	2 (3.57%)	1 (0.72%)	0.3380
HLA-B*38	6 (8.82%)	3 (5.36%)	17 (12.23%)	0.3274
HLA-B*39	2 (2.94%)	4 (7.14%)	5 (3.6%)	0.4482
HLA-B*40	4 (5.88%)	3 (5.36%)	10 (7.19%)	0.8719
HLA-B*41	1 (1.47%)	4 (7.14%)	7 (5.04%)	0.2981
HLA-B*44	10 (14.71%)	10 (17.86%)	19 (13.67%)	0.7575
HLA-B*47	4 (5.88%)	2 (3.57%)	6 (4.32%)	0.8115
HLA-B*48	0 (0%)	1 (1.79%)	2 (1.44%)	0.5767
HLA-B*49	4 (5.88%)	1 (1.79%)	6 (4.32%)	0.5222
HLA-B*50	3 (4.41%)	1 (1.79%)	5 (3.6%)	0.7160
HLA-B*51	7 (10.29%)	12 (21.43%)	19 (13.67%)	0.1994
HLA-B*52	4 (5.88%)	1 (1.79%)	7 (5.04%)	0.5129
HLA-B*53	2 (2.94%)	0 (0%)	2 (1.44%)	0.4092
**HLA-B*54**	3 (4.41%)	0 (0%)	0 (0%)	**0.0030**
HLA-B*55	2 (2.94%)	0 (0%)	4 (2.88%)	0.4356
HLA-B*56	3 (4.41%)	1 (1.79%)	1 (0.72%)	0.1880
HLA-B*57	1 (1.47%)	2 (3.57%)	6 (4.32%)	0.5701
HLA-B*58	3 (4.41%)	1 (1.79%)	1 (0.72%)	0.1880

Note: bold, *p*-values < 0.05 (chi-squared test).

**Table 4 ijms-26-07617-t004:** Comparative distribution of HLA-B alleles in HLA-B*27-negative SpA patients and control group 1 (HLA-SSP typing).

HLA-B	Control #1 (*n* = 335)	SpA (*n* = 263)	*p*-Value	RR [95% CI]
HLA-B*07	38 (11.34%)	37 (14.07%)	0.3228	1.240 [0.813–1.893]
HLA-B*08	52 (15.52%)	47 (17.87%)	0.5062	1.151 [0.803–1.650]
HLA-B*13	31 (9.25%)	28 (10.65%)	0.5832	1.150 [0.708–1.868]
HLA-B*14	15 (4.48%)	10 (3.8%)	0.8375	0.849 [0.388–1.860]
HLA-B*15	35 (10.45%)	19 (7.22%)	0.1968	0.692 [0.405–1.180]
HLA-B*18	74 (22.09%)	59 (22.43%)	0.9215	1.016 [0.751–1.373]
HLA-B*35	99 (29.55%)	75 (28.52%)	0.7865	0.965 [0.749–1.243]
HLA-B*37	8 (2.39%)	4 (1.52%)	0.5637	0.637 [0.194–2.093]
HLA-B*38	34 (10.15%)	26 (9.89%)	1.0000	0.974 [0.600–1.581]
HLA-B*39	16 (4.78%)	11 (4.18%)	0.8435	0.876 [0.413–1.855]
HLA-B*40	32 (9.55%)	17 (6.46%)	0.1802	0.677 [0.384–1.192]
HLA-B*41	9 (2.69%)	12 (4.56%)	0.2647	1.698 [0.726–3.971]
HLA-B*44	62 (18.51%)	39 (14.83%)	0.2716	0.801 [0.555–1.156]
**HLA-B*47**	4 (1.19%)	12 (4.56%)	**0.0189**	3.821 [1.246–11.720]
HLA-B*48	0 (0%)	3 (1.14%)	0.0845	-
HLA-B*49	16 (4.78%)	11 (4.18%)	0.8435	0.876 [0.413–1.855]
HLA-B*50	4 (1.19%)	9 (3.42%)	0.0886	2.866 [0.892–9.206]
HLA-B*51	61 (18.21%)	38 (14.45%)	0.2253	0.794 [0.547–1.151]
HLA-B*52	19 (5.67%)	12 (4.56%)	0.5823	0.805 [0.397–1.628]
**HLA-B*53**	0 (0%)	4 (1.52%)	**0.0369**	-
HLA-B*54	0 (0%)	2 (0.76%)	0.0845	-
HLA-B*55	10 (2.99%)	7 (2.66%)	0.7995	0.764 [0.281–2.076]
HLA-B*56	5 (1.49%)	5 (1.9%)	0.7553	1.274 [0.373–4.355]
HLA-B*57	8 (2.39%)	9 (3.42%)	0.4679	1.433 [0.560–3.664]
HLA-B*58	13 (3.88%)	5 (1.9%)	0.2278	0.490 [0.177–1.357]
HLA-B*73	2 (0.6%)	0 (0%)	0.5064	-

Note: Control #1 = control group 1; bold, *p*-values < 0.05 (chi-squared test).

**Table 5 ijms-26-07617-t005:** Comparative distribution of HLA-B alleles in HLA-B*27-negative SpA patients and control group 2 (HLA-NGS typing).

HLA-B	Control #2 (*n* = 1705)	SpA (*n* = 263)	*p*-Value	RR [95% CI]
HLA-B*07	188 (11.03%)	37 (14.07%)	0.1462	1.276 [0.919–1.771]
HLA-B*08	297 (17.42%)	47 (17.87%)	0.8616	1.026 [0.776–1.356]
HLA-B*13	138 (8.09%)	28 (10.65%)	0.1887	1.315 [0.895–1.933]
HLA-B*14	85 (4.99%)	10 (3.8%)	0.5354	0.763 [0.401–1.450]
HLA-B*15	149 (8.74%)	19 (7.22%)	0.4774	0.827 [0.522–1.309]
HLA-B*18	331 (19.41%)	59 (22.43%)	0.2460	1.156 [0.905–1.476]
HLA-B*35	444 (26.04%)	75 (28.52%)	0.4085	1.095 [0.889–1.348]
HLA-B*37	32 (1.88%)	4 (1.52%)	1.000	0.814 [0.289–2.273]
HLA-B*38	144 (8.45%)	26 (9.89%)	0.4114	1.171 [0.787–1.741]
HLA-B*39	88 (5.16%)	11 (4.18%)	0.6486	0.810 [0.438–1.496]
**HLA-B*40**	185 (10.85%)	17 (6.46%)	**0.0287**	0.596 [0.369–0.962]
HLA-B*41	70 (4.11%)	12 (4.56%)	0.7397	1.111 [0.611–2.022]
HLA-B*44	321 (18.83%)	39 (14.83%)	0.1236	0.787 [0.580–1.070]
**HLA-B*47**	21 (1.23%)	12 (4.56%)	**0.0007**	3.705 [1.844–7.441]
HLA-B*48	7 (0.41%)	3 (1.14%)	0.1387	2.778 [0.723–10.68]
HLA-B*49	51 (2.99%)	11 (4.18%)	0.3400	1.396 [0.738–2.648]
HLA-B*50	52 (3.05%)	9 (3.42%)	0.7031	1.122 [0.560–2.250]
HLA-B*51	316 (18.53%)	38 (14.45%)	0.1204	0.777 [0.572–1.063]
HLA-B*52	99 (5.81%)	12 (4.56%)	0.4751	0.786 [0.438–1.411]
HLA-B*53	11 (0.59%)	4 (1.52%)	0.1292	2.357 [0.756–7.351]
**HLA-B*54**	1 (0.06%)	2 (0.76%)	**0.0085**	19.45 [2.029–186.4]
HLA-B*55	52 (3.05%)	7 (2.66%)	0.6940	0.748 [0.325–1.724]
HLA-B*56	38 (2.23%)	5 (1.9%)	1.000	0.853 [0.339–2.148]
HLA-B*57	59 (3.46%)	9 (3.42%)	1.000	0.989 [0.496–1.970]
HLA-B*58	50 (2.93%)	5 (1.9%)	0.4253	0.648 [0.261–1.624]
HLA-B*73	2 (0.12%)	0 (0%)	1.000	-

Note: Control #2 = control group 2; bold, *p*-values < 0.05 (chi-squared test).

**Table 6 ijms-26-07617-t006:** Comparative distribution of HLA-B alleles in HLA-B*27-negative SpA patients and combined control population (*n* = 2040).

HLA-B	Control #3 (*n* = 2040)	SpA (*n* = 263)	*p*-Value	RR [95% CI]
HLA-B*07	226 (11.08%)	37 (14.07%)	0.1462	1.276 [0.919–1.771]
HLA-B*08	349 (17.11%)	47 (17.87%)	0.8616	1.026 [0.776–1.356]
HLA-B*13	169 (8.28%)	28 (10.65%)	0.1887	1.315 [0.895–1.933]
HLA-B*14	100 (4.9%)	10 (3.8%)	0.5354	0.763 [0.401–1.450]
HLA-B*15	184 (9.02%)	19 (7.22%)	0.4774	0.827 [0.522–1.309]
HLA-B*18	405 (19.85%)	59 (22.43%)	0.2460	1.156 [0.905–1.476]
HLA-B*35	543 (26.62%)	75 (28.52%)	0.4085	1.095 [0.889–1.348]
HLA-B*37	40 (1.96%)	4 (1.52%)	1.000	0.814 [0.289–2.273]
HLA-B*38	178 (8.73%)	26 (9.89%)	0.4114	1.171 [0.787–1.741]
HLA-B*39	104 (5.1%)	11 (4.18%)	0.6486	0.810 [0.438–1.496]
HLA-B*40	217 (10.64%)	17 (6.46%)	0.0287	0.596 [0.369–0.962]
HLA-B*41	79 (3.87%)	12 (4.56%)	0.7397	1.111 [0.611–2.022]
HLA-B*44	383 (18.77%)	39 (14.83%)	0.1236	0.787 [0.580–1.070]
**HLA-B*47**	25 (1.23%)	12 (4.56%)	**0.0007**	3.705 [1.844–7.441]
HLA-B*48	7 (0.34%)	3 (1.14%)	0.1387	2.778 [0.723–10.68]
HLA-B*49	67 (3.28%)	11 (4.18%)	0.3400	1.396 [0.738–2.648]
HLA-B*50	56 (2.75%)	9 (3.42%)	0.7031	1.122 [0.560–2.250]
HLA-B*51	377 (18.48%)	38 (14.45%)	0.1204	0.777 [0.572–1.063]
HLA-B*52	118 (5.78%)	12 (4.56%)	0.4751	0.786 [0.438–1.411]
HLA-B*53	11 (0.54%)	4 (1.52%)	0.1292	2.357 [0.756–7.351]
**HLA-B*54**	1 (0.05%)	2 (0.76%)	**0.0013**	23.27 [2.428–223.0]
HLA-B*55	62 (3.04%)	7 (2.66%)	0.6243	0.751 [0.328–1.719]
HLA-B*56	43 (2.11%)	5 (1.9%)	1.000	0.853 [0.339–2.148]
HLA-B*57	67 (3.28%)	9 (3.42%)	1.000	0.989 [0.496–1.970]
HLA-B*58	63 (3.09%)	5 (1.9%)	0.4253	0.648 [0.261–1.624]
HLA-B*73	4 (0.2%)	0 (0%)	1.000	-

Note: Control #3 = control group 3; bold, *p*-values < 0.05 (chi-squared test).

**Table 7 ijms-26-07617-t007:** Reported associations of non-HLA-B27 HLA-B alleles with spondylarthritis across diverse populations, and comparison with findings from the Romanian B27-negative cohort.

HLA-B	SpA Associations (Independent of HLA-B*27)	Findings from the Romanian Cohort (Our Study)
B*07	axSpA in French (*p* = 0.0008, OR = 5.91) [15]	No statistical association with SpA (*p* = 0.146)
	ReA in Americans (28.5% of the B*27-negative)	
	Idiopathic Sacroiliitis in Americans (50% of the B27 negative) [9]	
	AS in Black American (*p* < 0.025) [16]	
	USpA in Tunisian (*p* = 0.043, OR = 5.15) [10], Brazilian (*p* = 0.043, OR = 5.15) [11], Indian (*p* = 1.14 × 10^−7^, OR = 5.348) [12]	
B*13	AS in European (*p* = 0.00429, OR 1.43) [13]	No statistical association with SpA (*p* = 0.1887)
	PsA in Chinese (*p* = 4.0 × 10^−6^, OR 2.65) [17]	
B*14	Spondylarthritis in French (2 members of the same family out of 20 families) [18]	No statistical association with SpA (*p* = 0.5354)
	AS in Caucasians of European descent (*p* = 0.08, OR = 1.59) [14], Togolese (*p* = 0.0005, OR = 69) [19]	
B*15	USpA in Belgian (48%; *p* < 0.001) [20], Mexican (*p*<0.01, OR = 3.77) [21], Tunisian (*p* = 0.002, OR = 18.40) [10] Colombian (*p* < 0.0001, OR = 20.85) [22]	No statistical association with SpA (*p* = 0.1887)
B*18	PsA in Croatians [23] and Serbian [24]	No statistical association with SpA (*p* = 0.2460)
B*35	Increased risk of sacroiliitis (*p* = 0.021, OR = 6, 95% CI = 1.3–27), systemic inflammation (*p* = 0.047, OR= 4.7, 95% CI = 1–11.9), and peripheral arthritis (*p* = 0.003, OR = 5, 95% CI = 1.75–14.3) in Croatians with USpA [25] SpA in French (*p* = 0.018) [26]	No statistical association with SpA (*p* = 0.4085)
B*38	AS in Caucasians of European descent (American, British, Australian) (*p* = 0.04, OR = 3.2) [14] AS in Spanish (*p* < 0.01, OR = 5.38) [27]	No statistical association with SpA (*p* = 0.4114)
	PsA in Argentinian (*p* = 0.03, OR = 2.95) [28], Canadian (*p* = 0.04) [29]	
B*39	As in Japanese (*p* = 0.01) [30]	No statistical association with SpA (*p* = 0.6486)
	PsA in Canadian (*p* = 0.03) [29]	
B*40	peripheral SpA in French [15]	Protective effect for SpA (*p* = 0.0287, RR = 1.315 [0.895–1.933])
	AS in British [31], East Asian–Taiwanese (*p* < 0.001, OR = 2.8) [32], in an entire cohort of Taiwanese, Chinese, Korean (*p* = 2.54 × 10^−4^, OR = 1.65) [33], Caucasians of European descent (*p* = 0.001, OR = 0.71) [14]	
B*49	AS in European, Asian, African (*p* = 0.03, OR = 2.36) [14]	No statistical association with SpA (*p* = 0.1887)
B*51	ReA in Tunisian (*p* = 0.015, OR = 4.91) [10] Behcet in the USA and northern Europe (15%) [34]	No statistical association with SpA (*p* = 0.1204)
B*51:01	AS in Korean [35], European [13]	
B*52	AS in Caucasians of European descent (American, British, Australian) (*p* = 0.006, OR = 2.85) [14]	No statistical association with SpA (*p* = 0.4751)
B*57:03	USpA in Zambian (*p* < 0.05; OR = 5.24) [36]	No statistical association with SpA (*p* = 1)
	PsA in Chinese Han (*p* = 5.8 × 10^−5^, OR = 20.10) [17]	

Note: While most studies report positive associations for HLA-B*07 with SpA subtypes, a protective effect was identified in Europeans (*p* = 5.4 × 10^6^, OR = 0.82) [13], Han Chinese (*p* = 6 × 10^−4^, OR = 0.06), as well as African American populations (*p* = 0.048, OR = 0.46) [14].

**Table 8 ijms-26-07617-t008:** HLA-B genotypes and Hardy–Weinberg disequilibrium in HLA-B*27-negative SpA patients and combined control population (*n* = 2040).

	SpA			Control #3			
HLA*B Genotype	Observed Frequency (%)	Theoretical Probability (%)	D	Observed Frequency (%)	Theoretical Probability (%)	D	*p*-Value
B*08-B*18	**4.18**	1.18	**3.00**	**2.11**	0.93	**1.18**	0.0606
B*07-B*35	**3.04**	1.06	**1.99**	1.72	0.83	0.89	0.2101
B*13-B*35	**3.04**	0.80	**2.24**	1.03	0.63	0.40	**0.0139**
B*35-B*44	**3.04**	1.17	**1.87**	**2.40**	1.48	0.92	0.6761
B*15-B*35	**2.66**	0.54	**2.12**	1.57	0.66	0.91	0.2988
B*18-B*51	**2.66**	0.91	**1.75**	**2.30**	1.02	**1.28**	0.8853
B*35-B*51	**2.28**	1.14	**1.14**	**2.35**	1.42	0.94	0.8853
B*07-B*08	1.90	0.70	**1.21**	1.13	0.50	0.63	0.4362
B*07-B*18	1.90	0.84	**1.06**	1.32	0.60	0.73	0.636
B*07-B*44	1.90	0.55	**1.35**	1.23	0.58	0.65	0.5349
B*13-B*18	1.90	0.64	**1.26**	1.08	0.45	0.62	0.3885
B*18-B*35	1.90	1.80	0.10	**2.55**	1.54	**1.01**	0.6704
B*44-B*51	1.90	0.59	**1.31**	**2.35**	0.98	**1.37**	0.8092
B*08-B*08	1.52	0.98	0.54	0.54	0.78	−0.24	0.1455
B*08-B*35	1.52	1.48	0.04	**2.75**	1.29	**1.46**	0.3334
B*08-B*51	1.52	0.75	0.77	1.76	0.86	0.91	0.9728
B*18-B*38	1.52	0.61	0.91	0.83	0.47	0.36	0.4476
B*18-B*39	1.52	0.27	**1.25**	0.54	0.27	0.27	0.1455
B*18-B*44	1.52	0.93	0.59	**2.11**	1.07	**1.04**	0.6878
B*35-B*35	1.52	2.26	−0.73	**2.60**	2.13	0.46	0.3968
B*35-B*40	1.52	0.57	0.95	1.42	0.80	0.62	0.8823
B*08-B*41	1.14	0.24	0.90	0.39	0.17	0.22	0.2372
B*08-B*44	1.14	0.77	0.37	1.72	0.90	0.82	0.6659
B*08-B*49	1.14	0.21	0.93	0.29	0.16	0.14	0.1221
B*35-B*52	1.14	0.37	0.77	0.88	0.44	0.45	0.944
B*35-B*53	1.14	0.11	**1.03**	0.15	0.04	0.11	**0.0197**
B*38-B*44	1.14	0.40	0.74	1.27	0.45	0.82	0.9119
B*38-B*51	1.14	0.39	0.75	1.18	0.43	0.74	0.7998
B*40-B*44	1.14	0.30	0.84	1.03	0.56	0.47	0.8766
B*44-B*47	1.14	0.18	0.96	0.10	0.06	0.04	**0.0066**
B*07-B*38	0.76	0.36	0.40	0.34	0.25	0.09	0.62
B*07-B*40	0.76	0.27	0.49	0.59	0.31	0.28	0.9336
B*08-B*13	0.76	0.53	0.23	0.83	0.38	0.45	0.811
B*08-B*15	0.76	0.36	0.40	0.78	0.40	0.38	0.7409
B*08-B*38	0.76	0.51	0.25	0.78	0.39	0.39	0.7409
B*13-B*40	0.76	0.20	0.56	0.44	0.24	0.20	0.8168
B*13-B*55	0.76	0.07	0.69	0.15	0.07	0.08	0.191
B*15-B*41	0.76	0.09	0.67	0.15	0.09	0.06	0.191
B*15-B*50	0.76	0.06	0.70	0.20	0.06	0.13	0.295
B*18-B*18	0.76	1.43	−0.67	1.18	1.11	0.07	0.7711
B*18-B*41	0.76	0.30	0.46	0.34	0.20	0.14	0.62
B*18-B*47	0.76	0.27	0.49	0.05	0.06	−0.02	**0.0355**
B*18-B*49	0.76	0.25	0.51	0.25	0.19	0.06	0.4044
B*18-B*57	0.76	0.23	0.53	0.20	0.17	0.02	0.295
B*35-B*38	0.76	0.77	−0.01	1.08	0.65	0.43	0.8766
B*35-B*41	0.76	0.37	0.39	0.88	0.28	0.60	0.8788
B*35-B*47	0.76	0.34	0.42	0.25	0.09	0.16	0.4044
B*35-B*49	0.76	0.31	0.45	0.49	0.26	0.23	0.9061
B*35-B*56	0.76	0.14	0.62	0.25	0.16	0.09	0.4044
B*38-B*41	0.76	0.13	0.63	0.15	0.09	0.06	0.191
B*38-B*47	0.76	0.12	0.64	0	0.03	−0.03	**0.0047**
B*40-B*50	0.76	0.07	0.70	0.29	0.08	0.22	0.5138
B*44-B*57	0.76	0.15	0.61	0.25	0.17	0.08	0.4044
B*47-B*51	0.76	0.17	0.59	0.15	0.06	0.09	0.191
B*51-B*51	0.76	0.58	0.18	0.93	0.94	−0.01	0.944
Number of individuals	*n* = 263			*n* = 2040			**<0.0001**
HW test	**SpA**			**Control #3**			
P (chi-square)	**<0.0001**			**<0.0001**			

Note: SpA, spondyloarthritis; Control #3 = control group 3; *n*, number of individuals; D, disequilibrium coefficient; HW, Hardy–Weinberg equilibrium test; bold, frequencies values ≥ 2%, D values ≥ 1 and *p*-values < 0.05.

**Table 9 ijms-26-07617-t009:** The HLA-B genotypes identified only in HLA-B*27-negative SpA Patients (*n* = 263) and not present in the combined control population (*n* = 2040).

	SpA				Control #3		
HLA*B Genotype	Observed Frequency (%)	Theoretical Probability (%)	D	SpA Subtype	Observed Frequency (%)	Theoretical Probability (%)	D
B*13-B*37	0.38	0.04	0.34	A	0.00	0.04	−0.04
B*13-B*52	0.38	0.13	0.25	A	0.00	0.13	−0.13
B*18-B*54	0.38	0.05	0.33	A	0.00	0.00	0.00
B*47-B*52	0.38	0.06	0.32	A	0.00	0.02	−0.02
B*38-B*47	0.76	0.12	0.64	A/M	0.00	0.03	−0.03
B*35-B*48	0.38	0.09	0.29	M	0.00	0.03	−0.03
B*52-B*53	0.38	0.02	0.36	M	0.00	0.01	−0.01
B*15-B*48	0.38	0.02	0.36	P	0.00	0.01	−0.01
B*37-B*41	0.38	0.02	0.36	P	0.00	0.02	−0.02
B*57-B*57	0.38	0.04	0.34	P	0.00	0.03	−0.03

Note: D, disequilibrium coefficient; A, axial; M, mixed; P, peripheral; *n*, number of individuals.

**Table 10 ijms-26-07617-t010:** Frequencies of HLA-B genotypes in HLA-B*27-negative SpA patients stratified by axial, peripheral, and mixed phenotypes.

HLA*B Genotype	A (%)	M (%)	P (%)	HLA*B Genotype	A (%)	HLA*B Genotype	M (%)	HLA*B Genotype	P (%)
B*08-B*18	1.52	1.90	0.76	B*18-B*49	0.76	B*13-B*18	1.90	B*07-B*13	0.38
B*15-B*35	1.14	0.76	0.76	B*07-B*51	0.38	B*35-B*35	1.52	B*07-B*14	0.38
B*07-B*35	0.76	1.52	0.76	B*07-B*54	0.38	B*35-B*52	1.14	B*07-B*58	0.38
B*13-B*35	0.76	0.76	1.52	B*07-B*55	0.38	B*07-B*38	0.76	B*08-B*14	0.38
B*07-B*08	0.76	0.76	0.38	B*08-B*52	0.38	B*07-B*40	0.76	B*08-B*50	0.38
B*07-B*18	0.76	0.76	0.38	B*13-B*37	0.38	B*13-B*40	0.76	B*13-B*15	0.38
B*44-B*51	0.76	0.38	0.76	B*13-B*52	0.38	B*15-B*41	0.76	B*13-B*44	0.38
B*18-B*38	0.76	0.38	0.38	B*13-B*58	0.38	B*15-B*50	0.76	B*14-B*51	0.38
B*18-B*51	0.38	1.52	0.76	B*14-B*35	0.38	B*18-B*41	0.76	B*15-B*39	0.38
B*35-B*51	0.38	1.14	0.76	B*14-B*39	0.38	B*35-B*49	0.76	B*15-B*48	0.38
B*08-B*08	0.38	0.76	0.38	B*14-B*50	0.38	B*38-B*41	0.76	B*37-B*41	0.38
B*08-B*35	0.38	0.76	0.38	B*18-B*54	0.38	B*51-B*51	0.76	B*37-B*51	0.38
B*35-B*40	0.38	0.76	0.38	B*18-B*55	0.38	B*07-B*15	0.38	B*39-B*51	0.38
B*38-B*44	0.38	0.38	0.38	B*35-B*50	0.38	B*07-B*39	0.38	B*40-B*40	0.38
B*35-B*44	1.14	1.90		B*35-B*58	0.38	B*07-B*49	0.38	B*40-B*51	0.38
B*07-B*44	0.76	1.14		B*38-B*49	0.38	B*07-B*52	0.38	B*41-B*44	0.38
B*35-B*53	0.76	0.38		B*39-B*40	0.38	B*08-B*39	0.38	B*44-B*49	0.38
B*18-B*35	0.38	1.52		B*39-B*52	0.38	B*08-B*57	0.38	B*44-B*52	0.38
B*08-B*49	0.38	0.76		B*40-B*56	0.38	B*13-B*14	0.38	B*57-B*57	0.38
B*38-B*51	0.38	0.76		B*47-B*52	0.38	B*13-B*38	0.38		
B*40-B*44	0.38	0.76		B*51-B*56	0.38	B*13-B*57	0.38		
B*44-B*47	0.38	0.76		B*56-B*58	0.38	B*14-B*38	0.38		
B*08-B*13	0.38	0.38				B*14-B*52	0.38		
B*13-B*55	0.38	0.38				B*14-B*57	0.38		
B*18-B*57	0.38	0.38				B*15-B*18	0.38		
B*35-B*38	0.38	0.38				B*15-B*38	0.38		
B*35-B*41	0.38	0.38				B*18-B*40	0.38		
B*35-B*47	0.38	0.38				B*18-B*50	0.38		
B*38-B*47	0.38	0.38				B*18-B*58	0.38		
B*40-B*50	0.38	0.38				B*35-B*48	0.38		
B*08-B*51		1.14	0.38			B*35-B*57	0.38		
B*18-B*44		1.14	0.38			B*37-B*49	0.38		
B*18-B*39		0.76	0.76			B*38-B*38	0.38		
B*08-B*44		0.76	0.38			B*38-B*39	0.38		
B*08-B*41		0.38	0.76			B*40-B*55	0.38		
B*08-B*15		0.38	0.38			B*44-B*55	0.38		
B*08-B*38		0.38	0.38			B*48-B*51	0.38		
B*18-B*18		0.38	0.38			B*50-B*51	0.38		
B*18-B*47		0.38	0.38			B*51-B*55	0.38		
B*35-B*56		0.38	0.38			B*52-B*52	0.38		
B*44-B*57		0.38	0.38			B*52-B*53	0.38		
B*47-B*51		0.38	0.38						

Note: A, axial; M, mixed; P, peripheral; *n*, number of individuals.

**Table 11 ijms-26-07617-t011:** HLA-B genotypes and Hardy–Weinberg disequilibrium in HLA-B*27-negative SpA stratified by radiological sacroiliitis grade.

	Radiological Sacroiliitis Grade ≥ 2			No/Sacroiliitis Grade 1		
HLA*B Genotype	Observed Frequency (%)	Theoretical Probability (%)	D	Observed Frequency (%)	Theoretical Probability (%)	D
B*08-B*18	**6.67**	1.39	**5.28**	2.89	1.08	1.81
B*40-B*44	**3.33**	0.32	**3.01**	0.00	0.28	−0.28
B*07-B*44	**3.33**	0.64	**2.69**	1.16	0.49	0.66
B*35-B*44	**3.33**	0.92	**2.41**	2.89	1.31	1.58
B*18-B*51	**3.33**	0.93	**2.41**	2.31	0.89	1.42
B*07-B*18	**3.33**	1.23	**2.10**	1.16	0.67	0.49
B*40-B*50	2.22	0.12	**2.10**	0.00	0.04	−0.04
B*08-B*41	2.22	0.22	**2.00**	0.58	0.26	0.32
B*35-B*47	2.22	0.28	1.94	0.00	0.37	−0.37
B*18-B*49	2.22	0.31	1.91	0.00	0.22	−0.22
B*07-B*40	2.22	0.40	1.83	0.00	0.21	−0.21
B*15-B*35	2.22	0.43	1.80	2.89	0.61	**2.28**
B*44-B*51	2.22	0.48	1.74	1.73	0.65	1.08
B*13-B*35	2.22	0.57	1.65	**3.47**	0.94	**2.53**
B*13-B*18	2.22	0.62	1.60	1.73	0.63	1.10
B*35-B*51	2.22	0.85	1.37	2.31	1.31	1.00
B*35-B*35	2.22	1.63	0.59	1.16	2.62	−1.46
Number of individuals	*n* = 90			*n* = 173		
HW test	**Sacroiliitis grade ≥ 2**			**No/ sacroiliitis grade 1**		
P (chi-square)	**<0.0001**			**<0.0001**		

Note: D, disequilibrium coefficient; *n*, number of individuals; HW, Hardy–Weinberg equilibrium test; bold, frequencies values ≥ 3%, and D values ≥ 2.

**Table 12 ijms-26-07617-t012:** HLA-B genotypes and Hardy–Weinberg disequilibrium in HLA-B*27-negative SpA stratified by the presence of inflammation (elevated CRP or ESR).

	Elevated CRP or ESR			Normal CRP and ESR		
HLA*B Genotype	Observed Frequency (%)	Theoretical Probability (%)	D	Observed Frequency (%)	Theoretical Probability (%)	D
B*35-B*51	**5.13**	1.79	**3.34**	1.08	0.91	0.17
B*13-B*35	**3.85**	1.07	**2.77**	2.70	0.69	**2.01**
B*08-B*18	**3.85**	1.17	**2.68**	4.32	1.19	**3.14**
B*18-B*51	**3.85**	1.17	**2.68**	2.16	0.80	1.36
B*35-B*44	**3.85**	1.19	**2.65**	2.70	1.13	1.57
B*18-B*41	2.56	0.23	**2.33**	0.00	0.32	−0.32
B*35-B*53	2.56	0.24	**2.33**	0.54	0.07	0.47
B*40-B*44	2.56	0.25	**2.32**	0.54	0.32	0.22
B*07-B*38	2.56	0.27	**2.29**	0.00	0.40	−0.40
B*35-B*49	2.56	0.36	**2.21**	0.00	0.29	−0.29
B*18-B*39	2.56	0.39	**2.17**	1.08	0.22	0.86
B*15-B*35	2.56	0.60	1.97	2.70	0.51	**2.19**
B*44-B*51	2.56	0.62	1.95	1.62	0.57	1.06
B*13-B*18	2.56	0.70	1.86	1.62	0.61	1.01
B*35-B*40	2.56	0.71	1.85	1.08	0.51	0.57
B*07-B*18	2.56	0.86	1.71	1.62	0.84	0.79
B*08-B*08	2.56	0.92	1.64	1.08	1.00	0.08
B*07-B*35	2.56	1.31	1.25	**3.24**	0.95	**2.29**
B*35-B*35	2.56	3.46	−0.89	1.08	1.83	−0.75
Number of individuals	*n* = 78			*n* = 185		
HW test	**Elevated CRP or ESR**			**Normal CRP and ESR**		
P (chi-square)	**<0.0001**			**<0.0001**		

Note: CRP, C-reactive protein; ESR, erythrocyte sedimentation rate; D, disequilibrium coefficient; *n*, number of individuals; HW, Hardy–Weinberg equilibrium test; bold, frequencies values ≥ 3%, and D values ≥ 2.

**Table 13 ijms-26-07617-t013:** Frequencies of MICA–HLA-B haplotypes in Caucasian populations as reported in www.allelefrequencies.net.

Population	MICA–HLA-B Haplotype	Frequency (%)
USA Caucasian MIC (*n* = 103) [55]	MICA*002-B*35	7.3
	MICA*007-B*27	2.6
	MICA*008-B*07	8.6
	MICA*008-B*08	10
	MICA*008-B*44	8.7
	MICA*009-B*35	3.1
USA Caucasian MIC pop2 (*n* = 242) [56]	MICA*002-B*35:01	5.3
	MICA*004-B*44:03	4.8
	MICA*008-B*07:02	12
	MICA*008-B*08:01	20.6
	MICA*008-B*40:01	5
	MICA*008-B*44:02	10.6
	MICA*010-B*15:01	4.7
	MICA*010-B*44:03	1.5

## Data Availability

Data is contained within the article or Appendix A.

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
