# Peer review of "Genetic Complexity in Spondyloarthritis: Contributions of HLA-B Alleles Beyond HLA-B*27 in Romanian Patients"

_ijms, 2025, doi:10.3390/ijms26157617_

Round 1
Reviewer 1 Report
Comments and Suggestions for Authors
Ruxandra et al have investigated HLA-B genotypes in HLA-B27 negative patients with SpA. the study addresses a serious knowledge gap and despite being underpowered for certain analysis, demands consideration. However, the manuscript needs improvement on one aspect:
- Authors need to mention both loci observed in HLA-B allele. Was it so that there were certain combinations seen more often than others? This should be done for both subject groups.
Reviewer 2 Report
Comments and Suggestions for Authors
This study conducts a comprehensive immunogenetic analysis of HLA-B alleles in HLA-B27-negative Romanian patients with spondyloarthritis (SpA), addressing a critical research gap in Eastern European populations. The research identifies novel risk associations and potential protective effects , though several limitations require attention to enhance the study's validity.
1. The manuscript should supplement ethics approval information and address statistical power concerns regarding small subgroups (e.g., reactive arthritis n=5, HLA-B*54 n=3) by either combining rare SpA subtypes or employing robust statistical methods like bootstrap resampling.
2.The study would benefit from mechanistic insights by comparing peptide-binding grooves of risk alleles with HLA-B*27 using existing structural data.
3.For the male-biased HLA-B*41 association (p=0.032), researchers should either establish clinical correlations with male-specific manifestations or apply multiple testing corrections.
4. To evaluate potential linkage disequilibrium confounders, the analysis could incorporate haplotype data from tools like PHASE or reference European LD patterns from 1000 Genomes.
5.Finally, exploring genotype-phenotype correlations through stratification by radiographic damage or analysis of available inflammatory markers (CRP/ESR) and treatment response data could strengthen clinical relevance.
Round 2
Reviewer 1 Report
Comments and Suggestions for Authors
Authors have revised the manuscript and it can be accepted in the present form.
Author Response
We sincerely thank the reviewer for the positive evaluation and recommendation for acceptance.
Reviewer 2 Report
Comments and Suggestions for Authors
ACCEPT
Author Response
We are very grateful to the reviewer for their supportive assessment and for recommending our manuscript for acceptance.